# Electrochemical Determination of the Drug Colchicine in Pharmaceutical and Βiological Samples Using a 3D-Printed Device

**DOI:** 10.3390/molecules28145539

**Published:** 2023-07-20

**Authors:** Maria Filopoulou, Giorgios Michail, Vasiliki Katseli, Anastasios Economou, Christos Kokkinos

**Affiliations:** Laboratory of Analytical Chemistry, Department of Chemistry, National and Kapodistrian University of Athens, 157 71 Athens, Greece; mari.filop3@gmail.com (M.F.); geomixe1@gmail.com (G.M.); lilikats0@gmail.com (V.K.); aeconomo@chem.uoa.gr (A.E.)

**Keywords:** 3D-printing, voltammetry, colchicine, urine, sensor

## Abstract

In this work, a simple, fast, and sensitive voltammetric method for the trace determination of the alkaloid drug colchicine (Colc) using a 3D-printed device is described. The electrochemical method was based on the adsorptive accumulation of the drug at a carbon-black polylactic acid (CB/PLA) working electrode, followed by voltammetric determination of the accumulated species. The plastic sensor was printed in a single step by a low-cost dual extruder 3D-printer and featured three CB/PLA electrodes (serving as working, reference, and counter electrodes) and a holder, printed from a non-conductive PLA filament. The electrochemical parameters that affected the response of the device towards Colc determination, such as accumulation time and potential, solution pH, and other variables, were optimized. Under the selected conditions, the oxidation current of Colc was proportional to the concentration of Colc, and its quantification was conducted in the concentration range of 0.6–2.2 μmol L^−1^ with a limit of detection of 0.11 μmol L^−1^ in phosphate buffer (pH 7.0). Both within-device and between-device reproducibility were lower than 9%, revealing satisfactory operational and fabrication reproducibility. Furthermore, the 3D-printed device was employed for the voltammetric determination of Colc in pharmaceutical tablets and in human urine with satisfactory results, justifying its suitability for low-cost routine analysis of Colc.

## 1. Introduction

Colchicine (Colc), N-[(7S)-1,2,3,10-tetramethoxy-9-oxo-5,6,7,9-tetrahydrobenzo(a) heptalen-7-yl] acetamide, is a member of the proto-alkaloids group, prepared from the dried corns and seeds of the liliaceous plant meadow saffron (*Colchicum autumnale* L.). Its chemical structure is shown in Figure 1A. Colc demonstrates a multimodal mechanism of action and continues to be at the center of very recent biomedical, clinical, and toxicological research. Colc is a very old drug used for the relief of joint pain and serves as a specific anti-inflammatory agent in acute attacks of gout by inhibiting the migration of leucocytes to inflammatory areas, thus interrupting the inflammatory response that sustains the acute attack. In general, Colc impairs the liberation of enzymes from lysosomes and so retards the development of inflammation [1,2]. It is also applied to polyarthritis associated with sarcoidosis, and it has been approved for the treatment of Behcet’s disease and familial Mediterranean fever. Colc can be used in gastroenterology, especially in conditions of cirrhosis, as it reduces the formation of fibrous tissue in the liver [3,4]. The ability of Colc to bind tubulins to inhibit mitosis has made it a promising candidate for the treatment of cancer. Although Colc is not used clinically to treat cancer due to its toxicity, research has shown that it produces anti-vascular effects, leading to a bigger reduction in blood flow in tumors than in normal tissues and its ability to overcome P-glycoprotein efflux pump-mediated multidrug resistance [5]. These features render Colc a model compound for the development of novel anticancer drugs with a better toxicological profile, many of which are currently undergoing clinical studies. Additionally, there are numerous studies dealing with the potential role of Colc as an HIV inhibitor to treat AIDS [6]. Last but not least, Colc has been used in patients with COVID-19 as an immunomodulatory drug. The use of Colc in COVID-19 relies on the NLRP3 inflammasome activation caused by viroporin E, which is a component of COVID-19 and has an inflammatory response. As Colc reduces the inflammasome of NLRP3, it has been suggested for application in infections caused by COVID-19. Nevertheless, Colc did not decrease mortality or the duration of hospitalization in contrast to standard care for patients who were affected by COVID-19. In addition, the published reports are insufficient to suggest Colc as a therapy in patients affected by COVID-19 [7].

Despite its pharmacological utility, Colc can cause toxicity and significant side effects. Doses of Colc between 0.5 and 0.8 mg kg^−1^ can cause poisoning, while doses higher than 0.8 mg kg^−1^ may be fatal. The metabolism of Colc takes place partly in the liver, and then it is excreted in the urine and feces. It has been demonstrated that a low concentration of Colc (1.2 mg per day) causes a reduction in pain and gout symptoms, whereas a high dose of Colc (4.8 mg over 6 h) can cause common side effects, such as gastrointestinal upset including nausea, diarrhea, vomiting, low blood cell count, and rhabdomyolysis, as well as bone marrow damage, anemia, and hair loss. Colc causes oxidative stress in animals, leading to cognitive impairment, and its therapeutic use has been linked to sporadic Alzheimer’s disease in humans [8,9,10]. 

It is obvious that the quantification of Colc content in pharmaceutical and biological samples is very important in order to monitor the treatment of patients and perform pharmacological studies. Several analytical methods have been reported for Colc quantification, such as high-performance liquid chromatography (HPLC), mass spectroscopy (MS), spectrophotometry, HPLC-MS, and HPLC-UV [11,12,13,14,15]. Despite the advantages of these techniques, they require time-consuming sample preparation procedures and expensive instrumentation. In the quest for rapid, simple, and cheap methods for Colc determination, electrochemical methods exhibit their predominance, offering miniaturized portable instrumentation and point-of-need devices with extremely low cost and practical use. Different working electrodes have already been reported for electrochemical determination of Colc, mainly via voltammetry, including the hanging mercury drop electrodes [16,17,18], graphite-based screen-printed electrodes [19], boron-doped diamond electrodes [20,21], gold electrodes [22,23,24], bare glassy carbon electrodes [25], and modified glassy carbon electrodes with poly(*o*-phenylenediamine)/single-wall carbon nanotubes [26], or with acetylene black–dihexadecyl hydrogen phosphate composite film [27], or with magnetic ionic liquid/CuO nanoparticles/carbon nanofibers [28], and also carbon paste electrodes modified with multiwall carbon nanotubes [29]. All these electrochemical methods of Colc quantification make use of separate conventional “large-size” electrodes and do not present any degree of miniaturization, integration, or ease of manufacturing. On the contrary, three-dimensional (3D) printing gives the opportunity for the in-house production of completed electrochemical small systems. Fused deposition modeling (FDM) is an advanced 3D printing procedure in which an electrochemical device is CAD-designed with open-source software and printed from thermoplastic filaments. The filaments are heated to a semi-molten state and extruded on a platform, where they solidify, forming the device. This digital fabrication procedure makes use of low-cost and portable printers and does not require laboratory facilities, provides great flexibility in the size and geometry of the printed devices, involves fast fabrication speed, uses low-cost filaments with different properties, and is environmentally friendly as it does not use chemicals and does not produce waste. Moreover, FDM involves e-transferability of the device, as the design file format can be sent through e-mail and printed on every 3D-printer [30,31,32,33,34,35,36]. 

In this work, we exploited the advantages of FDM in the fabrication of a miniaturized and fully integrated 3D-printed device, which was applied for the first time to the voltammetric determination of Colc. The 3D-printed device was printed in a single step using a dual extruder 3D printer, and it was composed of three electrodes (printed by a carbon black–polylactic acid (CB/PLA)) filament and a holder (printed by a PLA filament) (Figure 1B). The 3D-printed device can be considered a ready-to-use sensor as it integrates the working, counter, and reference electrodes (WE, CE, and RE, respectively) and does not require any modification step with materials for the voltammetric determination of Colc. The developed voltammetric method was successfully applied to the analysis of pharmaceutical tablets and human urine.

## 2. Results and Discussion

### 2.1. Electrochemical Behavior of Colchicine at the 3D-Printed Device

The electrochemical behavior of Colc on the surface of a 3D-printed working electrode (WE) was studied using cyclic voltammetry (CV) and differential pulse voltammetry (DPV) in 0.1 mol L^−1^ phosphate buffer (PB) pH (7.0). In addition, the electrochemical properties of the 3D-printed electrodes were examined through electrochemical impedance spectroscopy (EIS) and cyclic voltammetry in a 0.1 mol L^−1^ KCl solution containing K_3_[Fe(CN)_6_] /K_4_[Fe(CN)_6_] as a redox probe. As shown in the cyclic voltammograms (Figure 2A) and differential pulse voltammograms (Figure 2B), when the 3D-printed electrode was used as-printed (red traces in Figure 2), the response towards Colc was poor, while when the WE was electrochemically activated (black traces in Figure 2), the sensitivity of WE was significantly enhanced, and Colc gave rise to a well-defined oxidation peak at about +1.3 V. Also, the cyclic voltammograms (Figure 2A) revealed that the Colc oxidation was irreversible at the 3D-printed device, showing similar behavior to that of other carbon-based electrodes [21,26,27,28]. The electrochemical activation of the 3D-printed WE included anodic polarization of the WE at +1.8 V for 200 s, followed by cathodic polarization of the WE at −1.8 V for 200 s. It has been shown before that after anodic polarization of carbonaceous electrodes, graphene oxide and functional groups are electrogenerated in situ on the surface of the electrodes, while the subsequent cathodic polarization converts the electrochemically produced graphene oxide to electrochemically active reduced graphene oxide, improving further the electrochemical properties of the electrodes [37,38,39]. Moreover, it has already been demonstrated that the electrochemical treatment of 3D-printed PLA carbonaceous electrodes removes parts of the polymeric material from the electrode surface, resulting in the exposure of the active carbonaceous material on the electrode surface [38,39]. As depicted in Figure 2B, the electrochemical activation causes an enhancement of the WE response towards the oxidation of Colc of about 73% compared to that of the as-printed 3D-printed WE. The Nyquist plots of the as-printed and activated WEs are presented in Figure 2C. The Nyquist plot of the as-printed 3D-printed WE (red trace) suggested a high charge transfer resistance (R_ct_), which was due to its relatively low conductivity. On the other hand, at the electrochemically activated 3D-printed WE (black trace), the R_ct_ decreased significantly, suggesting that the charge transfer resistance of the electrode was lowered and the conductivity enhanced. Hence, the significantly reduced R_ct_ can be attributed to the electro-etching effect induced by the electrochemical activation, which provides a faster electron transfer rate compared to the as-printed WE [37]. This fact was also confirmed by cyclic voltammetric studies in a 0.1 mol L^−1^ KCl solution containing 10 mmol L^−1^ K_3_[Fe(CN)_6_]/K_4_[Fe(CN)_6_] (Figure 2D), which revealed a significant increase in cathodic and anodic peak currents of the ferro/ferri probe at the electrochemically activated 3D-printed WE [37,39].

### 2.2. Influence of the pH, the Accumulation Conditions, the Voltammetric Waveforms on the Determination of Colc

The electrochemical method for the determination of Colc was based on the adsorptive accumulation of the drug on the surface of the electrochemically activated 3D-printed electrode, followed by the anodic voltammetric determination of the accumulated species. The adsorptive voltammetric response was evaluated with respect to the pH of the working solution, the voltammetric waveform, and the accumulation time and potential.

The electrochemical responses of Colc at the 3D-printed CB/PLA activated electrode were examined in solutions covering the pH range from 1.0 to 9.0, using HCl solutions with pH 1.0 and 3.0 and 0.1 mol L^−1^ PB with pH 5.0, 7.0, and 9.0. As depicted in Figure 3A, the solution pH had a strong effect on the voltammetric response, and the experiment results showed that the highest oxidation peak current was obtained at 0.1 mol L^−1^ PB with pH 7. 0, which was finally selected. The oxidation peak current shifted to more negative potentials by increasing the pH of the supporting electrolyte, and the inset of Figure 3A shows the linear relationship between the anodic peak potential (E_pa_) and the pH with a slope of 0.053 (V/pH), which is close to the anticipated theoretical value of 0.059 for the electrochemical reaction of Colc, involving two protons and two electrons [28]. Moreover, the differential pulse (DP) and square wave (SW) waveforms were tested (Figure 3B). The DP mode presented a higher oxidation peak current than the SW mode, and thus DP was selected. The adsorption of Colc on the electrode surface was affected by the duration of the accumulation process. As depicted in Figure 3C, the longer the accumulation period, the more Colc was adsorbed on the electrode surface and the larger the oxidation peak current. Considering the sensitivity and the analysis time, an accumulation period of 120 s was selected as a satisfactory compromise. The dependence of the oxidation peak current of Colc on the accumulation potential was examined over the range from −1.4 to +0.6 V. Equal adsorption efficiency was observed over the entire tested potential range, showing that the accumulation potential had no influence on the oxidation peak current of Colc. Therefore, the electrochemical method for the voltammetric determination of Colc was selected to be carried out in 0.1 mol L^−1^ PB (pH 7.0), applying an accumulation potential of −1.2 V for 120 s, and using the DP waveform. 

### 2.3. Analytical Performance and Interferences 

Next, the analytical features of the method were evaluated. The calibration curve was constructed by the oxidation peak current of Colc, obtained under optimized experimental parameters (i.e., in 0.1 mol L^−1^ PB (pH 7.0) applying an accumulation potential of −1.2 V for 120 s), versus different concentrations of Colc. The DP voltammetric response at different Colc concentrations in the range 0.6–2.2 µmol L^−1^ and the respective calibration plot are shown in Figure 4. The oxidation peak height showed a linear concentration dependence in the examined concentration range, with R^2^ = 0.997. The limit of detection (LOD) was 0.11 µmol L^−1^ Colc and was calculated by the equation LOD = 3 sd/a (where sd was the standard deviation of the intercept of the calibration plot and a was the slope of the calibration plot). The LOD of the developed method utilizing the 3D-printed device was comparable to that obtained with other voltammetric methods utilizing unmodified carbon-based electrodes, such as graphite screen-printed electrodes, boron-doped diamond electrodes, and glassy carbon electrodes (ranging from 0.1 to 2.0 µmol L^−1^) [17,19,21]. The within-device reproducibility in terms of % relative standard deviation (%RSD) was 5.4%, and it was estimated by measuring the DP voltammetric responses of 1.6 µmol L^−1^ Colc via eight repeated measurements. The between-device reproducibility (in terms of % RSD at five different devices) was 8.2% at 1.6 µmol L^−1^ of Colc. These electroanalytical results revealed the significant repeatability of the proposed 3D-printed device for the determination of Colc.

The effect of various possible interfering species, which can be included in pharmaceutical and biological samples, was examined by the addition of the substances to a solution containing 1 µmol L^−1^ Colc (Figure 4C). Glucose, sucrose, urea, uric acid, lactic acid, and K^+^, Na^+^, Ca^2+^, NH_4_ ^+^, Cl^−^, PO_4_^3−^, and SO_4_^2−^ did not interfere at a concentration ratio of 1:100 (Colc solution: interference compound), revealing adequate selectivity of the presented electrochemical method for the determination of Colc in real samples.

### 2.4. Applications

In order to verify the accuracy of the presented electrochemical method, the electrochemically activated 3D-printed device was applied to the determination of Colc in pharmaceutical tablets and in spiked human urine. Colc is mainly excreted via urine and feces, and 10–20% of Colc remains unchanged in urine; thus, measuring the quantity of discarded Colc is crucial to the calculation of its bioavailability and other pharmaceutical characteristics. The preparation procedures for both samples are described in Section 3.3. In human urine and pharmaceutical tablets, the determination of Colc quantity was carried out by the method of standard additions. Particularly in the pharmaceutical tablets, the Colc content was determined as 1.01 ± 0.05 mg per tablet, and the respective recovery was 101 ± 5% (n = 3), while in the case of a urine sample spiked with Colc, the obtained recovery was 97 ± 6% (n = 3) (Figure 5). These recovery values indicated that the accuracy of the presented voltammetric method utilizing the 3D-printed device was satisfactory and confirmed its usefulness for Colc determination in practical samples.

## 3. Materials and Methods

### 3.1. Reagents and Apparatus

The transparent non-conductive filament was polylactic acid (PLA) from 3DEdge, while the conductive filament was a carbon-black-loaded PLA filament from Proto-Pasta. The diameter of each filament was 1.75 mm. All the other reagents were purchased from Sigma-Aldrich. The pharmaceutical tablets (commercial name: COLCHICINA/ACARPIA from Acarpia) contained an average of 1 mg of Colc and were obtained from a local drug store. Besides, the pharmaceutical tablets contained as excipients the following materials: povidone, sucrose, microcrystalline cellulose, sodium starch glycollate, magnesium stearate, and purified talc. For the preparation of the phosphate buffer (PB), the appropriate amounts of Na_2_HPO_4_ and NaH_2_PO_4_ were mixed, and the pH value was adjusted by the addition of a 0.1 mol L^−1^ solution of HCl or NaOH. The electrochemical experiments were conducted by the PalmSense4 potentiostat (Palm Sens, Houten, The Netherlands) using the PS Trace 5.5 software (Palm Sens). 

### 3.2. Fabrication of the 3D-Printed Device 

The Tinkercad software was used for the design of the device, and the file was saved in .STL format. Then, the Flashprint software was used to open the file and to set the conditions of the 3D printing process, which were: 60 °C for the platform, 200 °C for the head dispensers for the printing of both PLA filaments (conductive and non-conductive) and a printing speed of 40 mm s^−1^. The respective file was saved in .3x format and transferred to an SD card. Next, the SD card was inserted in the appropriate slot of the Creator Pro dual extruder 3D-printer (Flashforge), and the printing process was started. In Figure 1A, a photograph of the 3D-printed device is displayed.

### 3.3. Electrochemical Measurements and Samples Analysis

The connection of the 3D-printed device to the portable potentiostat was accomplished with crocodile clips. For the differential pulse voltammetric measurements, the solution was stirred at 1000 rpm and a potential of −1.2 V for 120 s was applied to the working electrode, followed by a differential pulse scan in a static solution, and the voltammogram was recorded. The differential pulse parameters were: modulation amplitude, 50 mV; increment, 10 mV; pulse width, 75 ms; and pulse repeat time, 50 ms. The cyclic voltammograms were obtained in 0.1 mol L^−1^ PB (pH 7.0) at a scan rate of 50 mV s^−1^. The EIS studies were performed using a 0.1 mol L^−1^ KCl solution containing 25 mmol L^−1^ K_3_[Fe(CN)_6_]/K_4_[Fe(CN)_6_] as the electrochemical probe. The frequency ranged from 10^5^ to 1 Hz with the application of 0.1 V of potential (DC) and 0.01 V of amplitude (AC). For the differential pulse voltammetric analysis of pharmaceutical tablets, 4 tablets of COLCHICINA/ACARPIA (1 mg Colc per tablet) were pulverized and dissolved in 100 mL of doubly distilled water. Next, 100 µL of the resulting solution was transferred into the electrochemical cell, which had previously been filled with 9.9 mL of 0.1 mol L^−1^ PB (pH 7.0). The urine sample was obtained from a healthy 22-year-old male volunteer with his informed written consent, and the measurements were conducted in accordance with GCP regulations. The urine sample was spiked with Colc to a final concentration of 10 µmol L^−1^ Colc, and then it was diluted 1:10 with 0.1 mol L^−1^ PB (pH 7.0). In human urine and pharmaceutical tablets, the determination of Colc was carried out by the method of standard additions to minimize matrix effects. For the interference study, possible substances that can be present in pharmaceutical tablets and human urine, such as glucose, sucrose, urea, uric acid, lactic acid, and K^+^, Na^+^, Ca^2+^, NH_4_ ^+^, Cl^−^, PO_4_^3−^, and SO_4_^2−^ were added at a concentration of 100 µmol L^−1^ in 0.1 mol L^−1^ PB (pH 7.0) solution that contained 1 µmol L^−1^ Colc. All potentials at the 3D-printed device are referred to with respect to the carbon-black-loaded PLA reference electrode.

## 4. Conclusions

In summary, a 3D-printed device consisting of three carbon-black-loaded PLA electrodes was successfully applied to the DPV quantification of the drug Colc. The electrochemical device exploited the advantages of fused deposition modeling in terms of fabrication speed and operation simplicity, while electrochemical activation via anodic and cathodic polarization significantly improved the electrochemical response of the device towards Colc. The voltammetric method offered excellent sensitivity (LOD of 0.11 μmol L^−1^) and reproducibility (%RSD < 9%) and allowed the determination of Colc in real samples with satisfactory selectivity and without the requirement of any separation or complex sample pretreatment. These features made the proposed method of utilizing the 3D-printed device practical for the simple and low-cost routine determination of Colc.

## Figures and Tables

**Figure 1 molecules-28-05539-f001:**
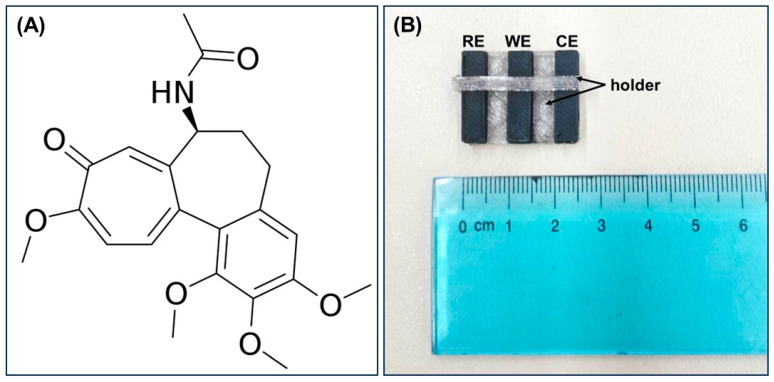
(**A**) Chemical structure of colchicine; (**Β**) photograph of the 3D-printed device.

**Figure 2 molecules-28-05539-f002:**
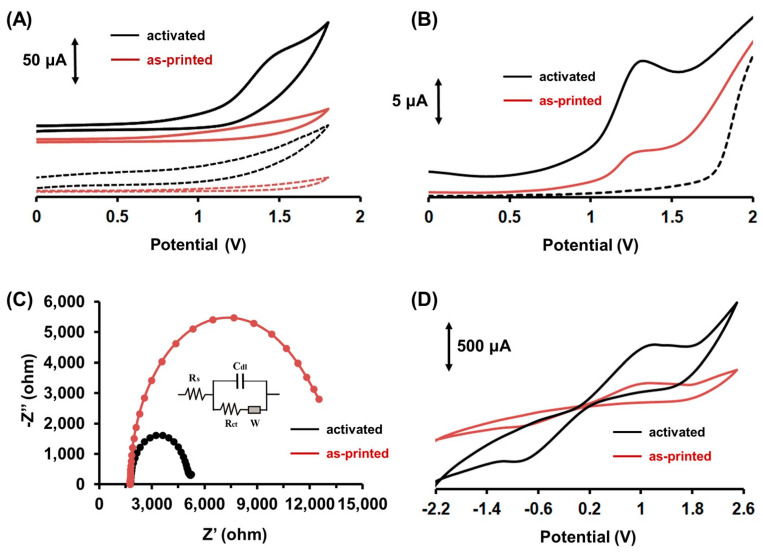
(**A**) Cyclic voltammograms in the absence (dot trace) and presence (solid trace) of 2 µmol L^−1^ Colc in 0.1 mol L^−1^ PB pH (7.0) at the as-printed 3D-printed CB/PLA electrode (red trace) and at the electrochemically activated 3D-printed CB/PLA electrode (black trace). (**Β**) Differential pulse voltammograms in the absence (dot trace) and presence (solid trace) of 1 µmol L^−1^ Colc in 0.1 mol L^−1^ PB (pH 7.0) at the as-printed 3D-printed CB/PLA electrode (red trace) and at the electrochemically activated 3D-printed CB/PLA electrode (black trace). Accumulation for 120 s at −1.2 V. (**C**) Nyquist plots of the as-printed 3D-printed CB/PLA electrode (red trace) and of the electrochemically activated 3D-printed CB/PLA electrode (black trace) in 0.1 mol L^−1^ KCl solution containing 25 mmol L^−1^ K_3_[Fe(CN)_6_]/K_4_[Fe(CN)_6_]. (**D**) Cyclic voltammograms of the as-printed 3D-printed CB/PLA electrode (red trace) and of the electrochemically activated 3D-printed CB/PLA electrode (black trace) in the presence of 10 mmol L^−1^ K_3_[Fe(CN)_6_]/K_4_[Fe(CN)_6_] in 0.1 mol L^−1^ KCl.

**Figure 3 molecules-28-05539-f003:**
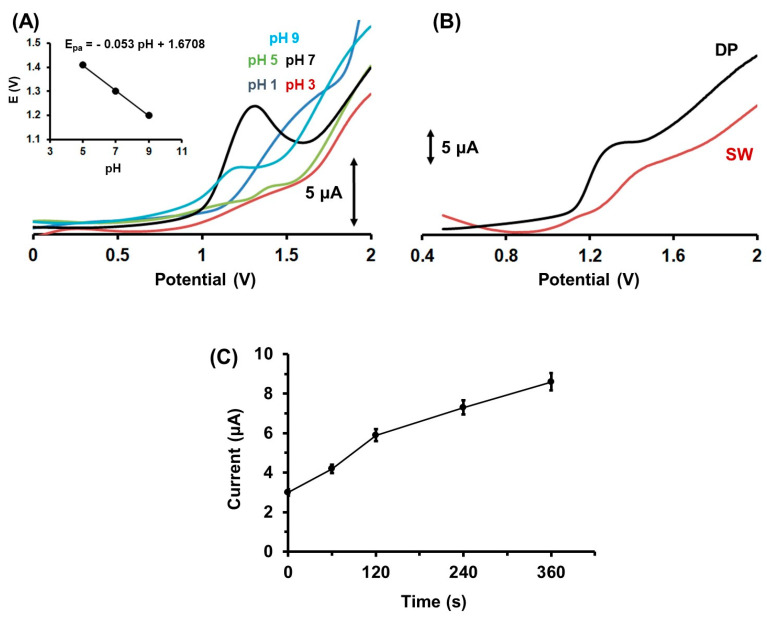
(**A**) Differential pulse voltammograms of 1 µmol L^−1^ Colc at an electrochemically activated 3D-printed CB/PLA electrode in solutions with different pH (HCl with pH 1.0 and 3.0, 0.1 mol L^−1^ PB with pH 5.0, 7.0 and 9.0). Inset: The linear relationship between the anodic peak potential of Colc oxidation and the pH. Accumulation for 120 s at −1.2 V. (**Β**) The effect of the scanning waveform (DP and SW) on the voltammetric response of 1 µmol L^−1^ Colc at electrochemically activated 3D-printed CB/PLA electrode. Accumulation for 120 s at −1.2 V. (**C**) The effect of accumulation time on the DPV response of 1 µmol L^−1^ Colc at electrochemically activated 3D-printed CB/PLA electrode. Error bars are the mean value ± sd (*n* = 3). Accumulation potential at −1.2 V.

**Figure 4 molecules-28-05539-f004:**
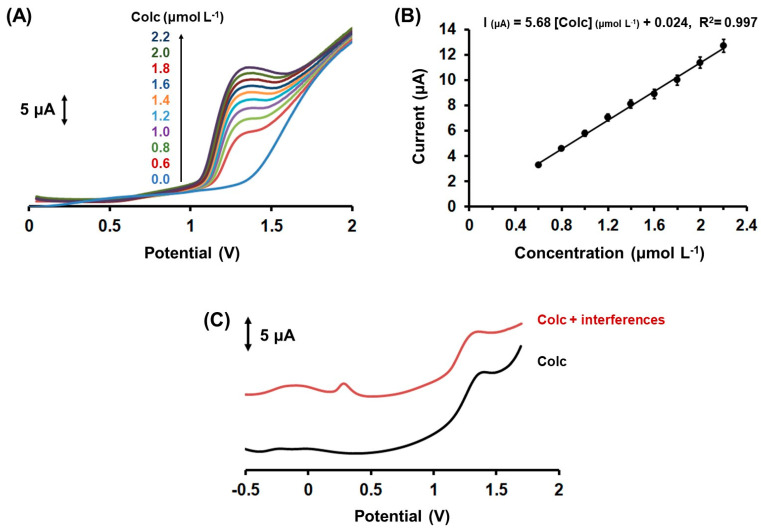
(**A**) Differential pulse voltammetric responses of Colc at 3D-printed CB/PLA sensor in 0.1 mol L^−1^ PB (pH 7) at different concentrations. From bottom to top: 0.0, 0.6, 0.8, 1.0, 1.2, 1.4, 1.6, 1.8, 2.0, and 2.2 µmol L^−1^ Colc. Accumulation for 120 s at −1.2 V. (**B**) The respective calibration plot. The points in the calibration plot are the mean value ± sd (n = 3). (**C**) The black trace is the DP voltammogram of 1 µmol L^−1^ Colc at the 3D-printed CB/PLA sensor in 0.1 mol L^−1^ PB (pH 7) and the red trace is the respective voltammogram of 1 µmol L^−1^ Colc in 0.1 mol L^−1^ PB (pH 7) containing 100 µmol L^−1^ of glucose, sucrose, urea, uric acid, lactic acid, and K^+^, Na^+^, Ca^2+^, NH_4_ ^+^, Cl^−^, PO_4_^3−^, and SO_4_^2−^. Accumulation for 120 s at −1.2 V.

**Figure 5 molecules-28-05539-f005:**
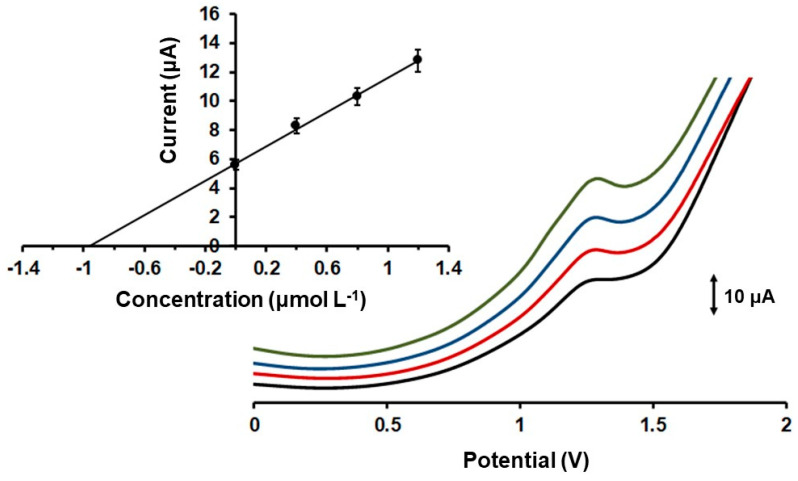
Differential pulse voltammograms obtained at the 3D-printed device for the determination of Colc in a human urine sample using the method of standard additions. Traces from below: urine sample spiked with Colc (black trace) and 3 standard additions of 0.4 µmol L^−1^ Colc (red, blue and green traces). Inset: the respective standard additions plot. The points in the plot are the mean value ± sd (n = 3). Accumulation potential −1.2 V for 120 s.

## Data Availability

All the used data have been provided in the text.

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
