# Peer review of "Electrochemical Determination of the Drug Colchicine in Pharmaceutical and Βiological Samples Using a 3D-Printed Device"

_molecules, 2023, doi:10.3390/molecules28145539_

Round 1

Reviewer 1 Report

 I recommend this paper to be published after minor revisions based on the following questions.

1. The advantages of 3D printing electrodes in the introduction should be added.

2. The impedance of the electrode needs to be tested.

3. The author needs to use a performance comparison between a traditional glassy carbon electrodes and the 3D printing electrodes in this experiment.

Reviewer 2 Report

The authors described the work of a simple, fast and sensitive voltammetric method for the trace determination of Colc using a 3D printed device, which showed its advantage for low-cost routine analysis of Colc. Some points should be carefully considered in the modification course.

(1) The authors may abbreviate the studied electrodes of as-printed 3D CB/PLA electrode and electrochemically activated 3D printed CB/PLA electrode into two short words for clear comparison.

(2) The response peak to Colc is at around 1.3V on the electrode, which may be close to the hydrogen evolution potential of the electrode. Could the authors provide the potential window of the electrochemical system and explain how to avoid the influence from hydrogen evolution?

(3) In the applications section (Line 204), the Colc concentration was 10 μmol L-1, but the listed detection range in Figure 4(b) was between 0.5~2.4 μmol L-1, so the tested range of the electrochemical system was not enough to illustrate its performance.

(4) In the conclusion section, some key parameters of the 3D printed electrode should be given including the detection efficiency.

Good for expression

Reviewer 3 Report

This work presents a rapid, simple, and sensitive electrochemical technique for the detection of the alkaloid drug colchicine. Although a simple 3D-printed device with an integrated working electrode, counter electrode, and reference electrode has been successfully developed, the work does not have sufficient experimental data to support the advantages of the prepared sensor and is somewhat thin. Nevertheless, this work is of some reference value. Overall, I believe that there are many weaknesses in this paper. The following points need to be addressed.

Q1. The author has written the article with a mix of abbreviations and full names. For example, in section 2.1, the abbreviations CV and DPV appear in the first sentence but are followed by "DP voltammogram", which is confusing.

Q2. The literature cited by the authors is quite old and it would be useful to provide the most recent published references.

Q3. When optimizing pH, the peak potential is usually shifted negatively as the pH increases, but Figure 3 (Α) shows that the peak potential at pH=3 is more negative than the peak potential at pH = 5.

Q4. The formatting of the graphs is confusing and it is recommended that the format be changed to be consistent and a legend added. Also, attention needs to be paid to cases and spaces between times and units.

Q5. There are significant grammatical issues throughout the article and it is recommended that this be revised.

There are significant grammatical issues throughout the article and it is recommended that this be revised.

Reviewer 4 Report

The article reports the employment of a 3D printed device for the electrochemical determination of Colchicine. Some electrochemical studies were performed to evaluate its analytical applicability. The article is well-written and organized and hence, I consider that it is suitable to be published in Molecules after taking into account some suggestions.

-As a general suggestion, the baseline substraction procedure could be useful in all voltammograms to identify clearly the peaks.

-Before the analytical application, the electrochemical characterization of the 3D-printed device could be performed to assess its electrochemical applicability; for example, the scan rate study in presence of an electrochemical redox probe, such as potassium ferrocyanide, may be carried out. The results obtained can be shown in the Supplementary material.

-Section 2.1. Including the cyclic and Differential Pulse voltammograms recorded in free-analyte solution in Figures 2A and 2B could be useful to ensure that the analytical signal is ascribed to the oxidation of the analyte. With respect to the cyclic voltammetry, some aspects regarding the irreversibility or reversibility of the process can be discussed, considering other pieces of research reported in literature.

-Section 2.2. In the pH study, the relationship between the oxidation peak and pH could be studied in order to estimate the numbers of protons and electrons involved in the electrochemical process. Both dependencies with the pH can be included as an inset of Figure 3A.

-Section 2.2, lines 159-164. In the main text, the effect of accumulation potential on the electrochemical response was reported, but no graph was included in the manuscript to illustrate it.

-Section 2.3. The linear regression curve and the corresponding adjusted coefficient (R2) could be included within Figure 4B.

-Section 2.3. The figures of merit obtained for the determination of colchicine, e.g. sensitivity and limit of detection, could be discussed and compared with those obtained with other electrochemical sensors reported in the literature.

-Section 2.3, line 192. I think that this part should be referred to Figure 4C. Furthermore, some chemical species mentioned in this section (Glucose, sucrose, ascorbic acid, K+, Na+, Ca2+, NH4+, Cl, PO43−, SO42−) did not appear in the figure caption 4C. On the other hand, uric acid was only included in this caption, while this compound was not mentioned in the main text.

-Section 2.4, lines 219-223. I understand that the urine sample was spiked at 10 µM and then, it was diluted ten times with the buffer solution (hence, the final spiked concentration of colc in the diluted sample is 1 µM). If I am right, the description of the caption could be confusing.

-Section 3.2. Maybe some details about the 3D printing process could be provided.

-Section 3.3. The interference study procedure can be detailed in this part. Furthermore, the electrochemical analysis of the real samples by using standard addition method could be explained. 

I believe that the quality of english language is acceptable. Maybe some minor changes could be carried out. For example, the revision of some expressions, such as ''duration of the accumulation time'' (line 154), instead of ''accumulation time'' and ''presented'' (line 152), instead of ''displayed'', among others.

Round 2

Reviewer 2 Report

The authors revised the manuscript carefully. It is recommened for publication even several tiny points do not affect the quality of this paper.

The language is OK.

Reviewer 3 Report

This work presents a rapid, simple, and sensitive electrochemical technique for the detection of the alkaloid drug colchicine. Although a simple 3D-printed device with an integrated working electrode, counter electrode, and reference electrode has been successfully developed, the work does not have sufficient experimental data to support the advantages of the prepared sensor and is somewhat thin. Nevertheless, this work is of some reference value. Overall, I believe that there are many weaknesses in this paper. the following points need to be addressed.

Q1. The literature cited by the authors is quite old and it would be useful to provide the most recent published references.

Q2. The formatting of the graphs is confusing and it is recommended that the format be changed to be consistent and a legend added. Also, attention needs to be paid to cases and spaces between times and units.

  Minor editing of English language required
